# Familial Arrhythmogenic Cardiomyopathy: Clinical Determinants of Phenotype Discordance and the Impact of Endurance Sports

**DOI:** 10.3390/jcm9113781

**Published:** 2020-11-23

**Authors:** Sarah Costa, Alessio Gasperetti, Argelia Medeiros-Domingo, Deniz Akdis, Corinna Brunckhorst, Ardan M. Saguner, Firat Duru

**Affiliations:** 1University Heart Center Zurich, Division of Cardiology, 8091 Zurich, Switzerland; sarah.costa@usz.ch (S.C.); alessio.gasperetti@usz.ch (A.G.); argelia.medeiros@hin.ch (A.M.-D.); deniz.akdis@usz.ch (D.A.); corinna.brunckhorst@usz.ch (C.B.); ardan.saguner@usz.ch (A.M.S.); 2Cardiogenetics—Swiss DNAlysis, 8600 Dübendorf, Switzerland; 3Center for Integrative Human Physiology, University of Zurich, 8057 Zurich, Switzerland

**Keywords:** arrhythmogenic cardiomyopathy, phenotype, genetics, exercise

## Abstract

Arrhythmogenic cardiomyopathy (ACM) is primarily a familial disease with autosomal dominant inheritance. Incomplete penetrance and variable expression are common, resulting in diverse clinical manifestations. Although recent studies on genotype–phenotype relationships have improved our understanding of the molecular mechanisms leading to the expression of the full-blown disease, the underlying genetic substrate and the clinical course of asymptomatic or oligo-symptomatic mutation carriers are still poorly understood. We aimed to analyze different phenotypic expression profiles of ACM in the context of the same familial genetic mutation by studying nine adult cases from four different families with four different familial variants (two plakophilin-2 and two desmoglein-2) from the Swiss Arrhythmogenic Right Ventricular Cardiomyopathy (ARVC) Registry. The affected individuals with the same genetic variants presented with highly variable phenotypes ranging from no disease or a classical, right-sided disease, to ACM with biventricular presentation. Moreover, some patients developed early-onset, electrically unstable disease whereas others with the same genetic variants presented with late-onset electrically stable disease. Despite differences in age, gender, underlying genotype, and other clinical characteristics, physical exercise has been observed as the common denominator in provoking an arrhythmic phenotype in these families.

## 1. Introduction

Arrhythmogenic cardiomyopathy (ACM) is a progressive, heritable cardiac condition with a highly variable phenotypic manifestation [1]. The classic form of ACM is arrhythmogenic right ventricular cardiomyopathy (ARVC), which is one of the leading causes of sudden cardiac death (SCD), especially in the young, athletic population. In up to 11% of index cases, SCD is the first disease expression, occurring even before manifestation of structural changes [2]. Approximately 60% of ARVC patients carry a genetic mutation [3], with the majority of genes coding for components of the desmosomes, which are essential for the structural integrity between cardiomyocytes [4], including plakophilin-2 (*PKP2*), plakoglobin (*JUP*), desmoglein-2 (*DSG2*), desmocollin-2 (*DSC2*) and desmoplakin (*DSP*). Inverted T waves and QRS prolongation in the right precordial leads (V1–V3) are among the common surface electrocardiogram (ECG) abnormalities typically seen in ARVC, accompanied by arrhythmias of right ventricular (RV) origin. Even though structural abnormalities are most dominant in the RV, left ventricular (LV) involvement may also occur with disease progression, and is rather common [5]. For example, while pathogenic variants in *PKP2* are usually associated with the classical form of ARVC [6], LV involvement can be observed in later stages of the disease. On the other hand, in the presence of *DSG2* or *DSC2* pathogenic variants, biventricular or LV-dominant phenotypes are quite common even in early disease [7].

Even though studies on the genotype–phenotype relationship in ARVC improved our understanding of the molecular mechanisms leading to the expression of the full-blown disease, the underlying genetic substrate and the clinical course of asymptomatic or oligo-symptomatic mutation carriers are still poorly understood. The aim of this case series was to analyze different phenotypic expression profiles of ARVC in the context of the same familial genetic mutation.

## 2. Experimental Section

Clinical characteristics of nine adult cases diagnosed with ARVC from the Swiss ARVC Registry belonging to four different families were analyzed. Demographic data, presenting complaints, physical findings, and data from 12-lead ECG, transthoracic echocardiography (TTE), cardiac magnetic resonance imaging (MRI), electrophysiological study (EP study), as well as treatment data were collected from the medical records. The genetic analyses were performed through next generation sequencing (NGS) in the probands using the TruSight cardio panel (Illumina), which contains 176 cardiac disease genes. Our analysis focused on cardiomyopathy genes: ABCC9, ACTC1, ACTN2, ANKRD1, BAG3, CAV3, CRYAB, CSRP3, DES, DMD, DSC2, DSG2, DSP, DTNA, EMD, FHL2, GLA, JUP, LAMA4, LAMP2, LMNA, LDB3, MYBPC3, MYH6, MYH7, MYL2, MYL3, MYLK2, NEXN, PKP2, PLN, PRKAG2, RBM20, SGCD, TAZ, TCAP, TGFB3, TMEM43, TMPO, TNNC1, TNNI3, TNNT2, TPM1, TTN, TTR, VCL. In family members, genetic analyses were performed through polymerase chain reaction (PCR) and Sanger Sequencing of single genes. The pathogenicity of genetic variants was assessed and confirmed by a clinical cardiogeneticist (AMD), according to the 2015 American College of Medical Genetics (ACMG) criteria [8], and modified by additional evidence from www.arvcdatabase.info and VarSome [9]. The study was approved by the Zurich Cantonal Ethical Committee (KEK-ZH-NR: 2014-0443). All patients signed a written informed consent for participation in this study.

## 3. Results

Figure 1 shows the family tree of all four families. Table 1 shows which of the current ARVC Task Force Criteria are fulfilled for each patient, while the imaging and ECG characteristics for all patients are shown in Table 2.

Family 1: this family with three cases harbors a pathogenic (P, Class V) heterozygous variant in *PKP2* (c.2146 − 1G > C).

The proband, a 56-year-old man, had been referred to our department initially at the age of 26, after experiencing multiple episodes of syncope. He was diagnosed with ARVC following TTE, RV angiography and EP study, during which a monomorphic ventricular tachycardia (VT) arising from the RV could be induced. He was implanted with a single-chamber implantable cardioverter defibrillator (ICD) at the age of 34. Based on the current 2010 Task Force Criteria, he has a diagnosis of definite ARVC (Table 1). His last TTE showed a heavily dilated RV with reduced function and a subtricuspid aneurysm, but no LV involvement. The patient underwent device replacements in 2003 and 2012, and is currently managed with amiodarone 100 mg once every other day and metoprolol 100 mg once daily. The patient used to be an endurance athlete (competitive soccer player) and still exercises at a moderate level at present time, despite recommendations to further reduce physical activity. The sister of the proband, a 65-year-old woman with the same genetic variants, who has never engaged in physical activity, remains completely asymptomatic up to date.

The niece of the proband, with a history of autoimmune hepatitis, was referred to our department in 2019 (at the age of 34) for evaluation of palpitations due to recurrent premature ventricular contractions (PVC) and dyspnea (New York Heart Association Class II). Based on the 2010 Task Force Criteria she has a diagnosis of definite ARVC with LV involvement (Table 1). Her last TTE showed a heavily dilated RV with reduced function and an apical aneurysm, and LV anteroseptal regional wall abnormalities. MRI showed a significantly dilated RV, depressed RV function with regional dyskinesia, microaneurysms, and late gadolinium enhancement (LGE) showing transmural fibrosis of the anteroseptal portion of the LV. EP study showed an inducible VT with RV inferobasal origin during isoprenalin provocation. Thus, a single-chamber ICD was implanted. The patient has been moderately physically active and is currently managed with metoprolol 25 mg daily.

Family 2: this family with two cases harbors a pathogenic (P, Class V) heterozygous variant in *PKP2* (c.1378 + 1G > C).

The proband, a 30-year-old man, was referred initially at the age of 19 after experiencing syncope during physical activity. His ECG showed PVCs of RV subtricuspid origin. An EP study showed inducibility of a monomorphic VT arising from the RV outflow tract (RVOT). Based on the 2010 Task Force Criteria, he was diagnosed with definite ARVC (Table 1). His last TTE showed a dilated RV with reduced function with a hypokinetic free wall and multiple subtricuspid aneurysms. MRI showed a heavily dilated RV with reduced function, regional dyskinesias and microaneurysms, as well as fibrosis of the RV free wall. The patient underwent implantation of a single-chamber ICD. He has been managed with sotalol 80 mg bid and is currently free of any further arrhythmias. This patient had always been strongly physically active performing endurance activities (competitive rowing) at maximum effort until disease diagnosis was made.

The mother of the proband, a 64-year old woman, was initially referred at the age of 58 because of the presence of the same pathogenic genetic variant of the proband on genetic cascade screening. No abnormalities were detected on ECG, TTE, MRI, and 48 h Holter. This year, the patient was referred again following detection of multiple non-sustained VT episodes on 24 h Holter. An EP study did not reveal any inducible arrhythmias. Since then, she has been managed with metoprolol 25 mg bid, and has not experienced any further arrhythmias. This patient had never been physically active.

The brother of the proband, a 35-year-old male, was referred initially at the age of 30 because of the presence of the same pathogenic genetic variant on genetic cascade screening. Upon complete cardiologic screening, no abnormalities were found. This patient had been moderately physically active.

Family 3: this family with two cases harbors a pathogenic (P, Class V) heterozygous variant in *DSG2* (c.523 + 2T > C).

The proband, a 26-year-old woman, was referred initially at the age of 18 because of the incidental ECG finding of T wave inversions in V1–V3. Based on the 2010 Task Force Criteria, she was diagnosed with definite ARVC (Table 1). Her last TTE showed a dilated RV with normal function and a subtricuspid aneurysm. An EP study was performed in 2013, which showed inducibility of a VT of superior axis with left bundle branch block (LBBB) morphology. Furthermore, RV angiography confirmed ARVC with the presence of a slightly dilated RV and inferoapical and RVOT aneurysms. She underwent implantation of a dual-chamber ICD and has been managed with metoprolol 25 mg bid since then. The patient used to perform endurance sports (long distance running), but she stopped competitive sports after ICD implantation, and has been moderately active since then.

The father of the proband, a 56-year-old man, was referred initially at the age of 52 for cascade screening and the presence of symptomatic PVCs of RV origin. Based on the 2010 Task Force Criteria, he was diagnosed with definite ARVC (Table 1). His last TTE showed a dilated RVOT with reduced function and a subtricuspid aneurysm. This patient had been moderately physically active. He refused implantation of an ICD and has since then been stable under therapy with nebivolol 2.5 mg daily.

Family 4: this family with two cases harbors a likely pathogenic (Class IV) heterozygous variant in *DSG2* (c.152G > C).

The proband, a 49-year-old woman, was referred initially at the age of 21 after experiencing syncope due to sustained VT. After diagnosis for definite ARVC (Table 1), she underwent implantation of a single-chamber ICD. Her last TTE showed a dilated RV with reduced function and multiple subtricuspid aneurysms, but no LV involvement. Furthermore, she had multiple VTs with left bundle branch block (LBBB) morphology and superior axis. At the age of 36, the patient refused ICD replacement, and since then, has been arrhythmia-free under metoprolol 25 mg daily. The patient used to be an endurance athlete (running and mountain biking), but has consistently diminished her physical activity after ARVC diagnosis.

The brother with no past medical history presented to the emergency department at the age of 53 with recurrent syncope due to sustained VT with right bundle branch block (RBBB) morphology and superior axis. TTE showed a normal, non-dilated LV with normal EF, a dilated right atrium and dilated RVOT (33 mm) with reduced RV function and a hypokinetic RV free wall. MRI showed regional dyskinesia and microaneurysms, as well as pronounced fibrosis and fatty infiltration of the RV free wall. The patient underwent implantation of a single-chamber ICD for secondary prevention, and since then, has been under bisoprolol 2.5 mg bid. The patient has always been moderately physically active.

## 4. Discussion

ACM is primarily a familial disease with autosomal dominant inheritance. Incomplete penetrance and variable expression are common, resulting in diverse clinical manifestations [10]. Furthermore, the clinical significance of the underlying genotype is unclear and its value in determining a clinical strategy has been controversial. The four different families described in our cohort demonstrate the challenges in clinical management of patients and their families with this disease.

### 4.1. Classical ARVC vs. Biventricular ACM

*PKP2* mutations account for a significant proportion of ACM cases, usually leading to ARVC, the classical RV dominant phenotype of ACM [11]. The clinical presentation is characterized by the presence of palpitations, exertional syncope, T wave inversions in the right precordial leads, ventricular arrhythmias with a LBBB pattern, and RV structural abnormalities on imaging in a young adult (20–40 years). ARVC is typically progressive, and after long-standing disease, end-stage RV as well as LV involvement may occur, leading to biventricular failure. In the biventricular variant of ACM, which is typically accompanied by a mutation in the desmosomal genes *DSP* or *DSG2*, there is involvement of both ventricles from early on. The disease shows regional RV dysfunction occurs along with apicolateral and baso-inferior fibrofatty infiltration in the LV, in the absence of reduced LV systolic function [12].

ACM patients may exhibit different phenotypes despite the same underlying genotype, as was the case in Family 1. The proband with *PKP2* genetic variant had a clear, RV-dominant ACM and a classical arrhythmic presentation, whereas his niece with the same genetic variant had a biventricular disease, with anteroseptal LV fibrosis and heart failure presentation. The proband had been very engaged in endurance sports, such as soccer and cycling, prior to his diagnosis. It is well known that excessive physical activity significantly contributes to early arrhythmic expression of disease, which explains the highly arrhythmic phenotype in this patient, whereas his affected sister who was never engaged in physical activity remained completely asymptomatic [13,14]. On the other hand, biventricular presentation of disease in the presence of a *PKP2* mutation, which was the case in the other affected family member of the proband, is rather unlikely [15]. In such cases, other confounding conditions such as inflammation may alter the clinical course of disease by acting as a precipitating factor. This particular patient suffered from autoimmune disease, which might have adversely contributed to disease progression. There has been growing evidence about the role of immune system in patients with ACM [16], but its potential role in our above-mentioned case may only be speculative. (Figure 2)

### 4.2. Early Onset ACM vs. No Disease

It is known that ACM probands usually present with more severe disease than the family members [17]. While this is clearly a disease with variable penetrance, it is still quite interesting to note how the presence of the same, clearly pathogenic, *PKP2* genetic variant in a family, can present with phenotypes, which are at the opposite end of the spectrum. The youngest family member (proband) in Family 2 who used to be an endurance athlete, as he was a rower on a competitive level, had an early-onset typical arrhythmic ARVC presentation whereas the much less athletic older brother and mother did not show any phenotypic manifestation of the disease even during long-term follow-up. (Figure 2) Clinical studies have shown that endurance athletes with desmosomal mutations were not only more likely to be at risk of life-threatening arrhythmias, but also more often developed the full ACM phenotype [13,14]. The RV of individuals with desmosomal mutations are particularly vulnerable to pathologic remodeling in response to exercise. Endurance exercise in these patients may facilitate myocyte uncoupling leading to fibrosis and adipocytosis as well as dysfunction of electrical coupling, which in turn facilitate the occurrence of arrhythmias [18]. This has been shown in a murine model by Kirchoff et al., using a *PKP2*-deficient mouse model, where endurance exercise accelerated the development of RV dysfunction and ventricular arrhythmias [19]. Furthermore, while variants in *PKP2* are relatively common in ACM patients, harboring one *PKP2* variant may not, by itself, be sufficient to provoke overt clinical disease. Concomitant causes such as environmental factors, which can be physical exercise, or a “second hit” in the same gene (compound heterozygosity) or in a second desmosome-encoding gene (digenic heterozygosity) have been shown to be required for development of the overt clinical phenotype. For this reason, other genes that encode proteins that either directly or indirectly contribute to the function and integrity of cellular junctions need to be screened [20].

### 4.3. Early and Unstable Electrical Phenotype vs. Late and Stable Phenotype

The patients in Families 3 and 4 were affected by two different heterozygous missense *DSG2* variants and demonstrated peculiar clinical phenotypes: an early unstable and highly arrhythmogenic electrical phenotype in the female probands (in their early twenties) and a late, stable, electrical phenotype in older male family members. In both families, physical exercise again seemed to play a dominant role in disease expression. The two female probands both used to be endurance athletes up until their diagnosis, and fortunately experienced electrical stabilization after significantly reducing their physical activity. These observations are in line with recent literature, which have suggested that detraining may reduce the arrhythmic burden in a cohort of athletes with ACM [21]. (Figure 3)

There is recent evidence that sex hormones may have an impact on susceptibility to occurrence of various arrhythmias. It has been shown that elite female athletes have significantly higher values of testosterone, when compared to controls [22]. Our group has recently shown that circulating sex hormone levels, particularly high levels of testosterone and low levels of estrogen, may lead to adverse arrhythmic outcomes in ARVC. This observation has also been confirmed in the experimental setting by using a stem-cell derived cardiomyocyte-based model [23]. The novel finding of increased ventricular arrhythmia susceptibility in patients with higher circulating levels of testosterone has also been verified in another and genetically distinct ARVC patient cohort [24].

In summary, in all four families described above, endurance sport was the most important uniting factor for phenotypic disease expression, independent of the underlying mutation. This is in line with previous observations, which showed that the amount and intensity of exercise increases the likelihood of adverse clinical outcomes in desmosomal mutation carriers [14].

## 5. Conclusions

Arrhythmogenic cardiomyopathy (ACM) is primarily a familial disease with autosomal dominant inheritance. Incomplete penetrance and variable expression are common, resulting in diverse clinical manifestations. Although recent studies on genotype–phenotype relationships have improved our understanding of the molecular mechanisms leading to the expression of the full-blown disease, the underlying genetic substrate and the clinical course of asymptomatic or oligo-symptomatic mutation carriers are still poorly understood. Despite differences in age, gender, underlying genotype, and other clinical characteristics, physical exercise has been observed as the common denominator in provoking an arrhythmic phenotype in ACM.

## Figures and Tables

**Figure 1 jcm-09-03781-f001:**
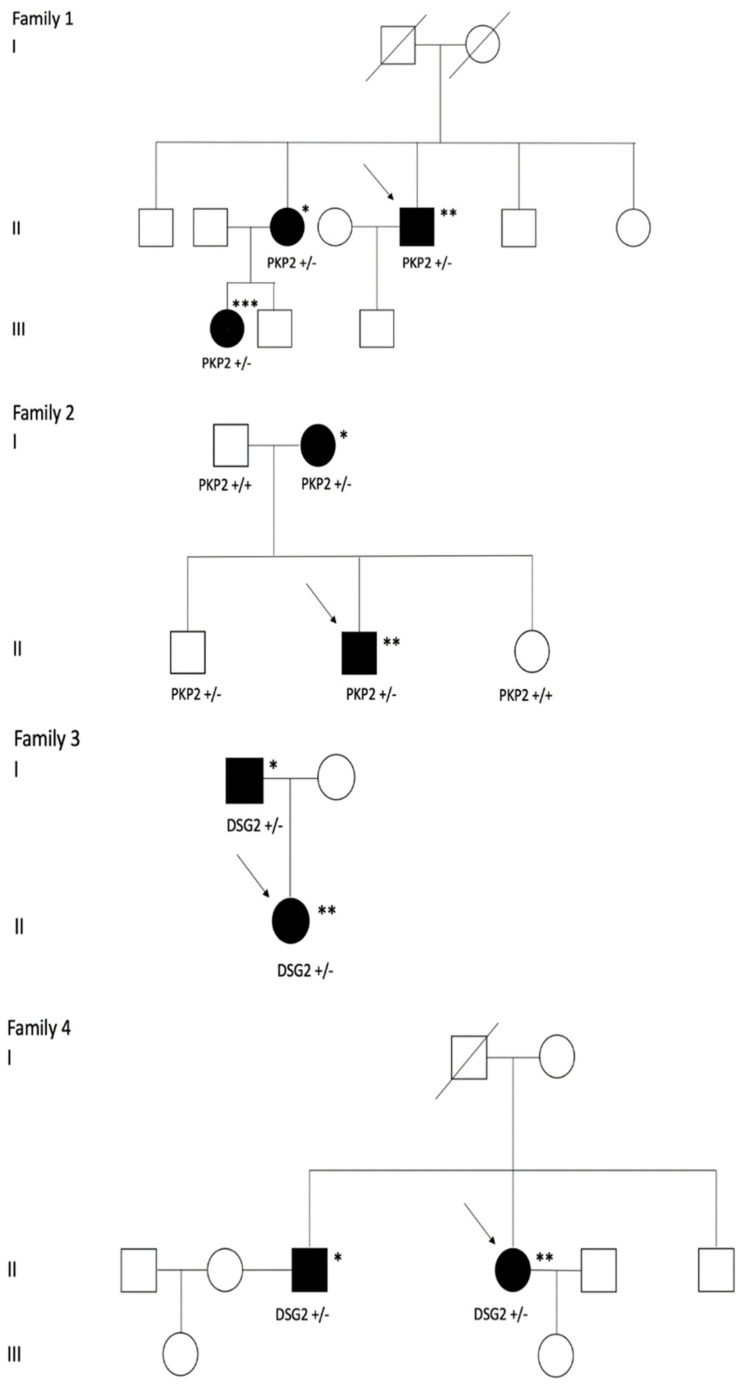
Family trees: Family 1: *PKP2* c.2146 − 1G > C; * asymptomatic; ** early, unstable electrical presentation; *** early, stable electrical, biventricular presentation. Family 2: PKP2 c.1378 + 1G > C; * asymptomatic; ** early, unstable electrical presentation. Family 3: DSG2 c.523 + 2T > C; * late, stable electrical presentation; ** early, unstable electrical presentation. Family 4: DSG2 c.294G > A; * late, stable electrical presentation; ** early, unstable electrical presentation. Arrow: Index patient.

**Figure 2 jcm-09-03781-f002:**
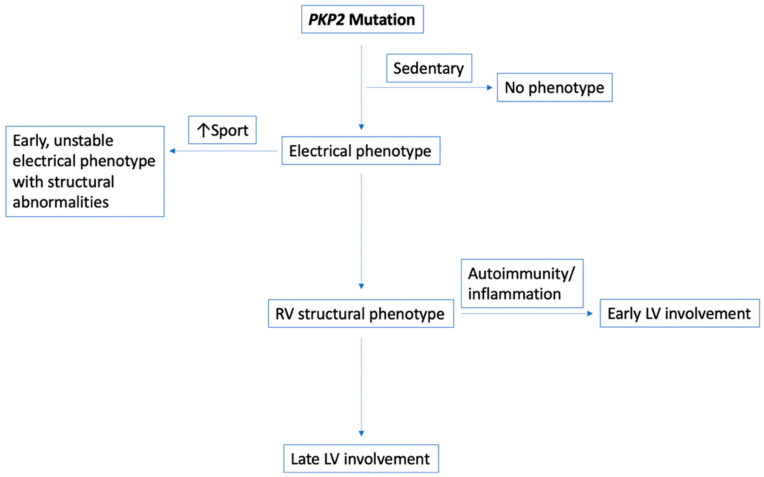
Genotype–phenotype determinants in subjects with a *PKP2* variant. The two families with a plakophilin-2 mutation present a broad phenotypic spectrum: from asymptomatic disease carrier to biventricular disease. While the clearest factor causing an early, unstable, and classic ARVC seems to be physical exercise, it interesting to note the autoimmune/inflammatory component in the presence of biventricular disease.

**Figure 3 jcm-09-03781-f003:**
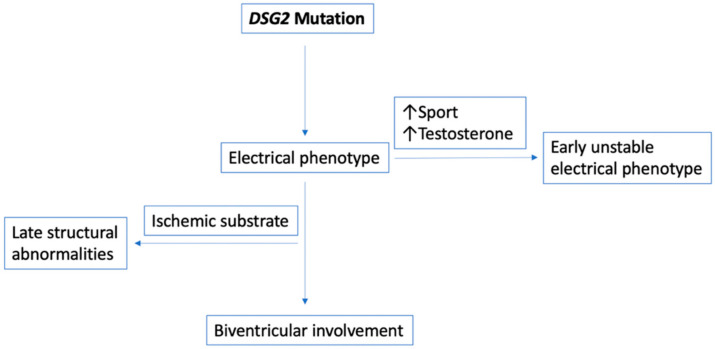
Genotype–phenotype determinants in subjects with a *DSG2* variant. The two families with a desmoglein-2 mutation present similarly: early, unstable involvement in the female, athletic proband and late involvement, partly precipitated by a concurrent ischemic substrate in the male, moderately active, family member.

**Table 1 jcm-09-03781-t001:** Swiss Arrhythmogenic Right Ventricular Cardiomyopathy (ARVC) Task Force Criteria.

Task Force Criteria														
		Structural		Histological		Repolarization		Conduction		Arrhythmias		Familial		TFC Score
		Major	Minor	Major	Minor	Major	Minor	Major	Minor	Major	Minor	Major	Minor	
Family 1	Proband	X		N/A	N/A	X		X			X *	X		9
	Niece	X		N/A	N/A	X					X *	X		7
Family 2	Proband	X		N/A	N/A	X					X	X		7
	Mother			N/A	N/A							X		2
	Brother			N/A	N/A							X		2
Family 3	Proband	X		N/A	N/A	X				X		X		7
	Father	X		N/A	N/A						X	X		5
Family 4	Proband	X		N/A	N/A	X				X		X		8
	Brother	X		N/A	N/A		X					X		5

X = Task force criterion fulfilled; N/A = no availability of investigation; * = based on 24h Holter premature ventricular contractions (PVC) count, no electrocardiogram (ECG) available for determination of ventricular tachycardia (VT) origin; TFC score: major criterion (2 points), minor criterion (1 point).

**Table 2 jcm-09-03781-t002:** Clinical Characteristics of Patients.

Patients Clinical Characteristics												
		Age at Diagnosis	Gender	Endurance Athlete	TTE (RV)			MRI			ECG	
					PLAX (cm/m^2^)	PSAX (cm/m^2^)	FAC (%)	RVEDVi (ml/m^2^)	RVEF (%)	LGE	T-Wave Inversions (Leads)	PVC Burden (%)
Family 1	Proband	26	M	Yes	2.9 *	2.6 *	26 *	N/A	N/A	N/A	III, aVR, V1-V4 *	2.5 *
	Niece	34	F	No	2.3 *	2.1 *	30 *	147 *	38 *	Yes *	II, III, aVF, V1-V6 *	1.5 *
Family 2	Proband	19	M	Yes	2.1 *	2.1 *	24 *	176 *	40 *	Yes *	II, III, aVF, V1-V3 *	3 *
	Mother	58	F	No	N/A	N/A	45	71	60	No	aVR, aVL	0
	Brother	30	M	No	N/A	N/A	52	93	60	No	aVR, V1	0
Family 3	Proband	18	F	Yes	1.8 *	1.87 *	44	97	57	No	aVR, V1-V3 *	3 *
	Father	52	M	No	2 *	1.8 *	35	79	62	No	aVR	13.8 *
Family 4	Proband	21	F	Yes	N/A	N/A	26 *	N/A	N/A	N/A	II, III, aVF, V1-V6 *	4 *
	Brother	53	M	No	N/A	N/A	28 *	81	61	Yes *	II, III, aVF, V5-V6 *	N/A

N/A = no availability of investigation; PLAX = parasternal long axis; PSAX = parasternal short axis; FAC = fractional area change; RVEDVi = right ventricular end diastolic volume (indexed); RVEF = right ventricular ejection fraction; LGE = late gadolinium enhancement; PVC = premature ventricular contractions measured by 24 h Holter; * = pathologic measurements giving the fulfillment of a major or minor Task Force Criterion.

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
