# Peer review of "Familial Arrhythmogenic Cardiomyopathy: Clinical Determinants of Phenotype Discordance and the Impact of Endurance Sports"

_jcm, 2020, doi:10.3390/jcm9113781_

Round 1

Reviewer 1 Report

The authors presented the case series of nine patients with established diagnosis of arrhythmogenic cardiomyopathy from four families. Two families had a pathogenic variant in gen coding the plakophilin-2 and other two families had a pathogenic variant in gen coding the desmoglein-2. The authors described the phenotype of each patient and analyzed the variability of phenotypes in the affected individuals with the same genetic variants from completely asymptomatic mutation carriers to patients with biventricular involvement and ventricular arrhythmias. The main observation of the authors was that unstable electrical phenotype was seen in patients engaged in intensive physical activity. The authors suggested that physical exercise is a dominator in provoking an arrhythmic phenotype in arrhythmogenic cardiomyopathy.

I have several comments.

  • May be I misunderstand something but I see ten cases described in the text: 3 cases in the Family 1, 3 cases in the Family 2, 2 cases in the Family 3 and 2 cases in the Family 4. In the abstract, in the experimental section, in the Tables there are nine cases mentioned.
  • In the experimental section it is worth to present the information about the genetic testing through NGS in more details: was it done using the target panel? How many gens were included in the panel? Were there only desmosomal gens or were there other priority gens known to be associated with the disease? What platform for sequencing was used?
  • Which of family members were screened: relatives of the first line, of the second line? How many relatives were screened in each family? Was medical examination clinically motivated or was performed in all family members? When was performed genetic testing (after clinical examination in the absence of phenotypical manifestations of the disease? or was performed in all family members independently from clinical results?).
  • The next comment is about the presentation of the Tables. It is difficult to read them when patients are coded 1,2 et.c. Please, include in the tables more specific information: proband, sister, brother et.c, also gender, age at the time of ARC diagnosis, level of physical activity. Please, check the number of Tables (line 66). Also, I would advise to calculate ARC diagnostic score for each patient and add it to the table (Kikuchi N et al, Long-term prognostic role of the diagnostic criteria for Arrhythmogenic right ventricular cardiomyopathy/dysplasia, 2016). It seems to me that it would better demonstrate the severity of the disease manifestation.
  • I have not found in the text any comments to Figures 2, 3. The detailed description should be performed.
  • In the Table with imaging and ECG data, please, specify PVC burden (how it was determined).
  • Line 270 – misprinting: “the same underlying phenotype”- should be genotype.
  • Line 322 – should be reference.

Reviewer 2 Report

Costa et al. provide a description of four families diagnosed with arrhythmogenic cardiomyopathy, two carrying mutations in PKP2 gene and two in DSG2 gene.

Clinical presentation with special attention to the task criteria is reported. The authors also attempt to correlate diseases severity with endurance physical exercise.

The study is well written and the subject is timely and the study relevant to the field.

I have some comments that could improve the form and clarity of the manuscript.

  1. The sentence “We aimed to analyze different phenotypic expression profiles of ACM in the context of the same familial genetic mutation” might be misleading and the reader could think the same mutation was analysed in all the families. I would suggest to explain a bit more and also mention the genes (PKP2 and DSG2) - if not also the specific mutations -,already in the abstract.
  2. Paragraph 3.1. I would sound more logic to refer first to Figure 1 and then to the Tables.
  3. Figure 1 – Family 1 – the mutated gene is not indicated under the affected individuals (while in Families 2, 3 and 4 it is). Please add.
  4. Line 75 – it is reported that the defibrillator was implanted “22 years ago”. However, this does not locate in the time at which age the patient received the defibrillator, which instead is more informative and relevant information.
  5. Line 140 – same as above : fourteen years ago does not place the event in the line of the patient’s age
  6. Line 141: one year ago: so the patient at referral was 54 or 53 years old? Please double check throughout the manuscript that the time is correctly mentioned in relation to the patients’ age
  7. Was myocardial biopsy available for any of these patients? If yes this information and relative figures of staining should be added
  8. One general comment remains on the expectations raised by the title “… and the impact of endurance sport”. In fact, among all of the individuals positive for the genotype presented in this paper, only one was practicing endurance sport. Also, in family 4, there was no difference between the two affected individuals, both being “physically active at moderate levels” (line 149). I therefore wonder whether the title is correct and whether this conclusion should instead come from a more focused study in athletes.
  9. There is also discrepancy with what mentioned in the discussion (referred to families 3 and 4) – lines 312-313: “…In both families, physical exercise again seemed to play a dominant role in disease expression”. As a consequence, I wonder whether Figures 2 and 3 stem from the data presented in this report, or rather from a more general analsysis of the literature and case reports, which should be mentioned.

Round 2

Reviewer 1 Report

I would like to thank the authors for their response and manuscript improvement. I do not have  comments any more.